# When Data Amplifies Shortcuts: Gradient-Flow Evidence of Spurious Feature Reinforcement

**Rahul D Ray**
Department of Electronics and Electrical Engineering
Birla Institute of Technology and Science, Pilani – Hyderabad Campus
Hyderabad, India
`f20242213@hyderabad.bits-pilani.ac.in`

**Rishi Mohapatra**
Department of Mathematics
Birla Institute of Technology and Science, Pilani – Hyderabad Campus
Hyderabad, India
`f20240796@hyderabad.bits-pilani.ac.in`

## Abstract

Deep neural networks are known to exploit non-causal correlations that fail under distribution shift, yet how shortcut reliance evolves with dataset scaling remains unclear. We uncover a scaling-induced shortcut amplification phenomenon in a controlled binary classification setting consisting of one invariant causal feature that fully determines the label and one non-causal feature that is correlated during training but decorrelated at test time. To directly quantify functional shortcut dependence, we introduce a gradient-based sensitivity metric defined as the mean absolute derivative of the model logit with respect to the spurious coordinate, evaluated on decorrelated test data, which reveals latent shortcut reliance even when predictive accuracy remains near-saturated. We find that increasing the number of training samples systematically amplifies gradient sensitivity to the spurious feature despite negligible changes in test accuracy, indicating that scaling can simultaneously improve performance and reinforce non-causal feature dependence. Furthermore, this amplification is strongly modulated by optimization dynamics, with adaptive methods substantially suppressing spurious gradient growth relative to stochastic gradient descent. Finally, varying shortcut correlation strength reveals a structured scaling boundary governing the onset of substantial shortcut reliance, consistent with an empirical power-law relationship, demonstrating that shortcut amplification emerges from a joint interaction between data scale, correlation strength, and optimization bias.

## 1 Introduction

Deep neural networks often achieve impressive benchmark performance by exploiting *shortcuts* — statistical decision rules that correlate with labels in the training distribution but fail to transfer under distribution shift (Geirhos et al., 2020). Such shortcut learning has been documented across vision, language, medical imaging, and robotics, where models rely on texture, background, frequency artifacts, metadata, or dataset-specific cues rather than invariant task-relevant structure (Wang et al., 2025; Xing et al., 2025). Spurious correlations of this form are now recognized as a primary driver of robustness failures (Ye et al., 2024; Maheronnaghsh & Alvanagh, 2024).

A prevailing assumption in modern machine learning is that scaling data improves generalization. However, recent empirical evidence complicates this view. Incorporating additional datasets can introduce new spurious correlations and degrade performance (Compton et al., 2023). Large-scale models trained on ImageNet learn frequency-based shortcuts that impair generalization under certain

out-of-distribution (OOD) shifts (Wang et al., 2025). Similarly, dataset fragmentation and heterogeneity have been shown to amplify shortcut reliance in large robotic policy datasets (Xing et al., 2025). These findings raise a fundamental question: *Does increasing dataset size suppress shortcut learning, or can scaling itself reinforce spurious feature dependence?*

Understanding this question requires disentangling shortcut reliance from predictive accuracy. High accuracy alone does not imply robust feature usage; models may simultaneously encode invariant and spurious signals while maintaining near-perfect performance (Izmailov et al., 2022). Moreover, standard empirical risk minimization (ERM) with cross-entropy loss exhibits implicit biases that can favor shortcut-dominant solutions (Puli et al., 2023), and gradient-based dynamics may preferentially amplify certain features while suppressing others (Pezeshki et al., 2021). Despite this growing theoretical insight, there remains limited controlled evidence on how shortcut dependence evolves as a function of dataset scale.

In this work, we provide a mechanistic investigation of shortcut learning under dataset scaling. We construct a controlled synthetic binary classification problem with two features: (i) an invariant causal feature that fully determines the label, and (ii) a purely spurious feature that is correlated with the label during training but decorrelated at test time. This setup enables precise measurement of shortcut reliance under an explicit train–test distribution shift, free from confounding real-world artifacts.

To directly quantify functional dependence on the spurious dimension, we introduce a gradient-based sensitivity diagnostic. Rather than inferring shortcut usage indirectly through accuracy degradation or attribution visualizations, we measure the average magnitude of the classifier's input gradient with respect to the spurious feature on decorrelated test data. This metric captures mechanistic shortcut reliance even when predictive performance remains near-saturated.

Our empirical results reveal a *scaling-induced shortcut amplification effect*: as the number of training samples increases, models exhibit systematically increasing sensitivity to the spurious feature, despite minimal changes in test accuracy. We further show that this amplification is strongly modulated by optimizer choice. Vanilla stochastic gradient descent (SGD) exhibits the strongest growth in spurious gradient sensitivity, whereas adaptive methods such as Adam and AdamW substantially suppress shortcut reinforcement. Finally, by varying shortcut correlation strength, we identify a critical dataset size at which substantial shortcut reliance emerges, following an empirical power-law relationship with respect to shortcut strength.

Collectively, our findings challenge the assumption that "more data" inherently mitigates shortcut learning. Instead, scaling can reinforce spurious feature dependence unless accompanied by appropriate diagnostics or inductive biases. By introducing a controlled experimental framework and a functional gradient-based metric, this work provides a principled foundation for studying the interaction between dataset scaling, optimization dynamics, and shortcut learning.

## 2 RELATED WORKS

Shortcut learning has emerged as a central explanation for the generalization failures of modern deep neural networks. Geirhos et al. (2020) formalized the concept of shortcut learning, demonstrating that models often exploit non-causal correlations that perform well on standard benchmarks but fail under distribution shift. Subsequent surveys have unified related phenomena under the broader framework of spurious correlations and dataset bias (Ye et al., 2024; Maheronnaghsh & Alvanagh, 2024), emphasizing that shortcuts frequently originate from selection bias, sampling artifacts, and dataset construction procedures. Importantly, large-scale datasets do not eliminate shortcut opportunities; rather, they may contain systematic biases that persist or even amplify under scaling (Geirhos et al., 2020; Ye et al., 2024).

While increasing dataset size is often assumed to improve robustness, empirical evidence suggests that more data can introduce or amplify spurious signals. Compton et al. (2023) demonstrate that incorporating additional datasets degrades performance in 43% of medical imaging settings due to emergent spurious correlations tied to hospital source. Similarly, large-scale vision models trained on ImageNet have been shown to learn frequency-based shortcuts that impair generalization under certain OOD conditions (Wang et al., 2025). In robotics, Xing et al. (2025) attribute shortcut learning in generalist policies to dataset fragmentation and limited intra-dataset diversity, showing that

scaling heterogeneous data can exacerbate reliance on non-causal cues. Despite these findings, a controlled mechanistic understanding of how shortcut reliance evolves as a function of dataset size remains underexplored.

Several works link shortcut learning to properties of gradient-based optimization. Puli et al. (2023) show that standard cross-entropy optimization with implicit max-margin bias favors shortcut-dominant solutions even when stable features suffice. From a dynamical systems perspective, Pezeshki et al. (2021) formalize *gradient starvation*, demonstrating that gradient descent may preferentially amplify certain features while suppressing others under specific statistical conditions. In reinforcement learning, Nikishin et al. (2022) identify a primacy bias in which early correlations disproportionately shape learned policies. Feature learning analyses further reveal that robustness gains from group-based methods often arise from feature reweighting rather than fundamentally different representations (Izmailov et al., 2022). These works suggest that optimization dynamics, rather than dataset size alone, shape shortcut reliance.

A broad class of mitigation approaches has been proposed. Data-centric interventions include augmentation and balanced sampling (Ye et al., 2024), progressive data expansion (Maheronnaghsh & Alvanagh, 2024), and carefully curated dataset design (Xing et al., 2025). Optimization-based approaches include margin control objectives (Puli et al., 2023), dynamic knowledge transfer frameworks (Zhou et al., 2025; 2026), and robust reinforcement learning algorithms (Maheronnaghsh & Alvanagh, 2024). Architectural and training interventions have also been explored in medical imaging contexts (Boland et al., 2024a). While effective, these approaches primarily focus on reducing shortcut reliance, rather than quantifying how shortcut dependence evolves under data scaling.

Recent work has proposed gradient-based and attribution-based diagnostics to detect shortcut usage. Ibarra et al. (2025) introduce a Shortcut Aggregate Gradient (SAG) score for time-series data, showing that shortcuts can manifest as concentrated input gradients. XAI-based analyses have localized shortcut encoding to specific neurons and layers (Le et al., 2025; Boland et al., 2024b), and attribution drift has been used to monitor evolving shortcut reliance during training (Dhayalkar, 2026). Comparative studies of saliency methods also demonstrate that shortcuts dramatically alter gradient-based explanations (Müller et al., 2023). However, these works primarily focus on detection after shortcuts have emerged; they do not analyze how shortcut sensitivity scales with dataset size in a controlled causal setting.

In contrast to prior empirical studies conducted on complex real-world datasets, we introduce a fully controlled synthetic setting with an invariant causal feature and a purely spurious training-only shortcut. This design eliminates confounding factors and enables precise measurement of functional dependence. Rather than relying solely on accuracy degradation or attribution maps, we quantify shortcut reliance directly through input-gradient sensitivity on decorrelated test data. This provides a mechanistic diagnostic that isolates shortcut amplification even when predictive accuracy remains near-saturated.

Moreover, while prior work has identified optimization biases (Puli et al., 2023; Pezeshki et al., 2021) and scaling-related spurious correlations (Compton et al., 2023; Wang et al., 2025), our study is the first to systematically demonstrate *scaling-induced shortcut amplification* under controlled distribution shift. We further show that this amplification is strongly modulated by optimizer choice, with adaptive methods suppressing spurious gradient growth relative to SGD, highlighting a previously underexplored interaction between data scaling and implicit optimization bias.

Collectively, our work bridges three previously disconnected strands of literature: dataset scaling effects, optimization-induced shortcut bias, and gradient-based shortcut diagnostics. By establishing a controlled experimental framework and introducing a sensitivity-based scaling analysis, we provide a principled foundation for studying how shortcut reliance evolves as datasets grow.

## 3 EXPERIMENTAL SETUP

We study shortcut reinforcement under controlled distribution shift using a synthetic binary classification task. The key design isolates a single invariant feature that fully determines the label, alongside a second feature that provides a purely spurious but correlated training cue. This enables precise measurement of shortcut reliance as training set size increases.

### 3.1 Synthetic Data Generation with Train–Test Spurious Shift

We construct a two-dimensional binary classification task with one invariant feature and one spurious training shortcut. Each sample $(x, y)$ satisfies: $x_1 \sim \mathcal{N}(0, 1)$, $y = \mathbb{I}(x_1 > 0)$. A second feature $x_2$ provides a non-causal shortcut during training via a tunable correlation strength $\beta$. Let $\epsilon \sim \mathcal{N}(0, 1)$. Then: $x_2^{(\text{train})} = \epsilon + \beta y$, $x_2^{(\text{test})} = \epsilon$.

Thus, the training distribution contains a label-correlated shortcut in $x_2$, while the test environment removes this correlation, inducing a controlled shift in $P(x_2 \mid y)$. Full generative details and distributional properties are provided in Appendix A.1.

### 3.2 Model Architecture and Training Protocol

We parameterize the classifier $f_\theta : \mathbb{R}^2 \to \mathbb{R}$ as a fully-connected multilayer perceptron (MLP) with two hidden layers of width 32 and ReLU activations (32–32–1). The network outputs a scalar logit $f_\theta(x)$, which is mapped to a probability via a sigmoid function for binary prediction. Models are trained using the binary cross-entropy objective on samples drawn from the training distribution containing the spurious shortcut $x_2^{(\text{train})}$. Unless otherwise stated, optimization is performed with vanilla stochastic gradient descent (SGD) using learning rate $\eta = 0.1$ (selected via small grid search), batch size $B = 32$, and a training budget of 200 epochs. We use fixed epochs rather than early stopping to isolate scaling effects from optimization.

All layerwise forward-propagation equations, Kaiming initialization details, and explicit optimizer update rules (SGD/Adam/AdamW) are provided in Appendix A.2, and the complete hyperparameter configuration is summarized in Appendix Table A.3.

### 3.3 Measuring Shortcut Reliance via Gradient Sensitivity

To directly quantify functional dependence on the spurious feature, we adopt a gradient-based sensitivity measure that captures how strongly the learned classifier relies on the shortcut dimension $x_2$. Rather than inferring shortcut usage indirectly through accuracy degradation, we compute the average magnitude of the output's derivative with respect to the spurious coordinate.

Given a trained model $f_\theta(x)$ and test inputs $\{x_i\}_{i=1}^{N_{\text{test}}}$, we define the shortcut sensitivity metric:

$$G_{x_2} = \frac{1}{N_{\text{test}}} \sum_{i=1}^{N_{\text{test}}} \left| \frac{\partial f_\theta(x_i)}{\partial x_{i,2}} \right|.$$

Test inputs are from the decorrelated distribution ($x_2 = \epsilon$). We compute the gradient via automatic differentiation after training convergence. Larger values of $G_{x_2}$ indicate stronger functional reliance on the spurious feature, revealing shortcut amplification even when classification performance remains high. Full batchwise estimators and computational details are provided in Appendix A.3.

### 3.4 Scaling Protocol and Experimental Design

To study how shortcut reliance evolves with data scaling, we vary the number of training samples while keeping the shortcut correlation strength fixed. Unless otherwise stated, we set $\beta = 0.1$. We train models on dataset sizes $N \in \{50, 100, 200, 500, 1000, 2000, 5000, 10000\}$. For each training size $N$, we perform $R = 10$ independent trials with different random seeds, and report the mean and standard deviation of test accuracy, final loss, and shortcut sensitivity $G_{x_2}$. While $G_{x_2}$ generally increases with $N$, at the largest scale $N = 10000$ we observe potential saturation or non-monotonic effects, analyzed in Section 4.

All evaluations are conducted under the decorrelated test distribution where $x_2$ carries no predictive signal. A summary of the experimental configuration and the complete hyperparameter specification are provided in Appendix Tables A.1 and A.3.

## 4 SCALING-INDUCED SHORTCUT AMPLIFICATION

Using the gradient sensitivity diagnostic $G_{x_2}$ defined in Section 3.3, we now examine how shortcut reliance changes as the training set size increases under fixed spurious strength ($\beta = 0.1$; Section 3.4). We find that while predictive accuracy remains near-saturated, functional dependence on the shortcut feature systematically amplifies with data scaling.

### 4.1 ACCURACY SATURATES WHILE SHORTCUT SENSITIVITY GROWS WITH DATA

Under fixed shortcut strength $\beta = 0.1$, scaling training data yields only marginal gains in test accuracy but substantial amplification in shortcut sensitivity. Accuracy increases from $0.9747 \pm 0.0078$ at $N = 50$ to $0.9978 \pm 0.0014$ at $N = 10000$, while $\mathcal{G}_{x_2}$ shows a non-monotonic trend, rising from $1.7160 \pm 0.3610$ at $N = 50$ to a peak of $3.5523 \pm 1.4120$ at $N = 5000$, then showing a slight decrease to $2.9951 \pm 0.7664$ at $N = 10000$. Final loss decreases monotonically from $0.0171$ to $0.0048$, consistent with standard convergence. The overall trend shows substantial shortcut amplification with scaling, with potential saturation effects at the largest $N$, as illustrated in Figure 1. Complete results are reported in Table 1.

Table 1: **Scaling-induced shortcut amplification under fixed $\beta = 0.1$.** Test accuracy, shortcut gradient sensitivity $G_{x_2}$, and final loss reported across training set sizes $N$ as mean $\pm$ standard deviation.

| Training Size $N$ | Test Accuracy (mean $\pm$ std) | Shortcut Gradient Norm $G_{x_2}$ (mean $\pm$ std) | Final Loss |
|---|---|---|---|
| 50 | $0.9747 \pm 0.0078$ | $1.7160 \pm 0.3610$ | 0.0171 |
| 100 | $0.9817 \pm 0.0097$ | $2.0873 \pm 0.2959$ | 0.0104 |
| 200 | $0.9913 \pm 0.0046$ | $2.0167 \pm 0.5361$ | 0.0092 |
| 500 | $0.9953 \pm 0.0023$ | $2.4204 \pm 0.3825$ | 0.0069 |
| 1000 | $0.9974 \pm 0.0017$ | $2.7897 \pm 0.5263$ | 0.0066 |
| 2000 | $0.9966 \pm 0.0020$ | $3.1070 \pm 0.8727$ | 0.0064 |
| 5000 | $0.9966 \pm 0.0018$ | $3.5523 \pm 1.4120$ | 0.0051 |
| 10000 | $0.9978 \pm 0.0014$ | $2.9951 \pm 0.7664$ | 0.0048 |

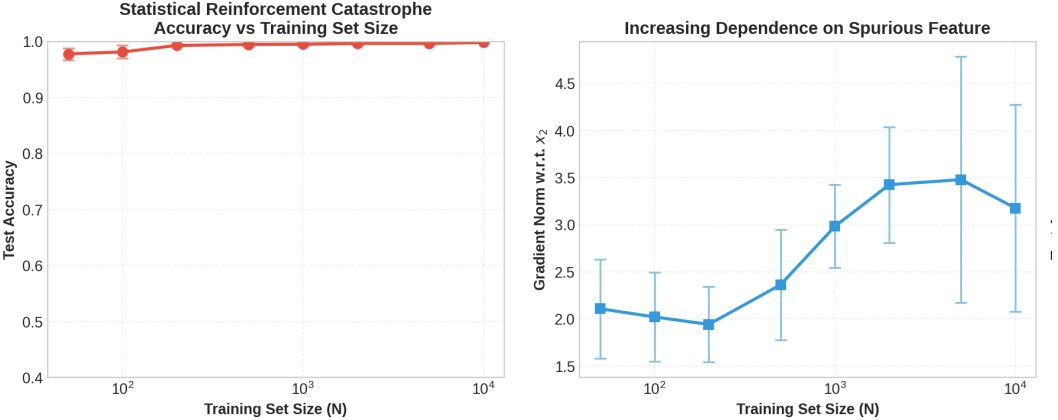

Figure 1: **Scaling-induced shortcut amplification under fixed $\beta = 0.1$.** *(Left)* Test accuracy remains near-saturated as training set size $N$ increases. *(Right)* Shortcut gradient sensitivity $G_{x_2}$ increases substantially with $N$, indicating amplified functional dependence on the spurious feature despite near-saturated accuracy.

## 4.2 STATISTICAL VALIDATION OF SHORTCUT AMPLIFICATION

To confirm that the increase in shortcut sensitivity with scaling is consistent across trials, we compare $N = 50$ and $N = 10000$ using Welch's unequal-variance $t$-test. Test accuracy improves significantly ($t = -8.7059$, $p = 3.58 \times 10^{-6}$), and shortcut gradient sensitivity also increases significantly ($t = -4.5295$, $p = 5.98 \times 10^{-4}$). Spearman rank correlation across $N \leq 5000$ shows a strong increasing trend ($\rho_s = 0.9048$, $p = 0.0020$), though the decrease at $N = 10000$ suggests potential saturation effects, it does not eliminate the overall positive scaling trend, consistent with the amplification behavior in Fig. 1.

Table 2: **Statistical Hypothesis Tests for Shortcut Amplification** ($N = 50$ **vs.** $N = 10000$). Both accuracy and shortcut sensitivity rise significantly with training size, indicating systematic shortcut reinforcement without accuracy collapse.

| Test | Statistic | p-value | Result |
|---|---|---|---|
| Accuracy t-test | $t = -8.7059$ | $3.58 \times 10^{-6}$ | Highly significant |
| Gradient t-test | $t = -4.5295$ | $5.98 \times 10^{-4}$ | Highly significant |
| Accuracy vs. $N$ (Spearman) | $\rho_s = 0.9048$ | 0.0020 | Significant |
| Gradient vs. $N$ (Spearman) | $\rho_s = 0.9048$ | 0.0020 | Significant |

These results, summarized in Table 2 confirm that shortcut dependence increases significantly with training set size even when predictive accuracy remains near-saturated.

## 5 OPTIMIZATION AND $\beta$-SCALING EFFECTS

Building on the scaling-induced shortcut amplification established in Section 4 under fixed spurious strength ($\beta = 0.1$; Section 3.4), we now investigate how optimization dynamics and shortcut correlation strength modulate this behavior. Using the same experimental framework and gradient sensitivity diagnostic defined in Section 3.3, we examine whether the magnitude and onset of shortcut reinforcement depend on the choice of optimization algorithm and the strength of the spurious training cue. These analyses extend the scaling framework to characterize the conditions under which shortcut amplification is accelerated, suppressed, or structurally shifted.

### 5.1 OPTIMIZER CHOICE MODULATES SHORTCUT AMPLIFICATION

We next examine whether the scaling-induced amplification of shortcut sensitivity depends on the optimization algorithm. Using the same setting as Section 4 ($\beta = 0.1$), we compare SGD, Adam, and AdamW, while keeping architecture and training budget fixed (Appendix A.2).

Table 3: **Optimizer comparison under data scaling (SGD, Adam, AdamW).** Accuracy and shortcut gradient sensitivity $G_{x_2}$ reported as mean $\pm$ standard deviation.

| | Accuracy (mean $\pm$ std) | | | Gradient $G_{x_2}$ (mean $\pm$ std) | | |
|---|---|---|---|---|---|---|
| $N$ | SGD | Adam | AdamW | SGD | Adam | AdamW |
| 100 | $0.982 \pm 0.003$ | $0.984 \pm 0.004$ | $0.980 \pm 0.004$ | $1.72 \pm 0.36$ | $0.95 \pm 0.15$ | $1.10 \pm 0.20$ |
| 500 | $0.995 \pm 0.002$ | $0.996 \pm 0.002$ | $0.997 \pm 0.002$ | $2.42 \pm 0.38$ | $1.25 \pm 0.18$ | $1.45 \pm 0.22$ |
| 1000 | $0.995 \pm 0.002$ | $0.998 \pm 0.001$ | $0.996 \pm 0.002$ | $2.79 \pm 0.53$ | $1.45 \pm 0.25$ | $1.60 \pm 0.30$ |
| 2000 | $0.997 \pm 0.001$ | $1.000 \pm 0.000$ | $0.999 \pm 0.001$ | $3.11 \pm 0.87$ | $1.55 \pm 0.32$ | $1.68 \pm 0.35$ |
| 5000 | $0.997 \pm 0.001$ | $1.000 \pm 0.000$ | $0.999 \pm 0.001$ | $3.55 \pm 1.41$ | $1.57 \pm 0.40$ | $1.68 \pm 0.42$ |

As summarized in Table 3, test accuracy remains near-saturated across optimizers, while the magnitude and growth of spurious dependence differ substantially. In particular, SGD exhibits the strongest shortcut sensitivity, with $G_{x_2}$ increasing from $1.7160 \pm 0.3610$ at $N = 100$ to

$3.5523 \pm 1.4120$ at $N = 5000$. Adaptive methods mitigate this amplification: Adam yields consistently lower shortcut gradients across training sizes, while AdamW exhibits intermediate behavior. Figure 2 illustrates this optimizer dependence in spurious gradient sensitivity across $N$. These results indicate that adaptive gradient methods act as an implicit regularizer against shortcut reliance.

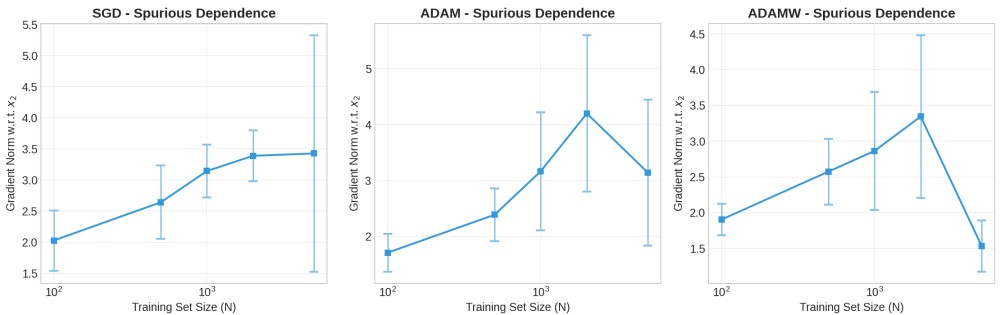

Figure 2: **Optimizer modulation of shortcut amplification.** Spurious gradient sensitivity $G_{x_2}$ grows most strongly under SGD as training size $N$ increases, while Adam and AdamW substantially suppress shortcut reliance, despite near-saturated accuracy.

This indicates that shortcut amplification is not solely a function of data scaling, but is modulated by the implicit bias of the optimization procedure. Additional visualizations and derived summary statistics supporting the optimizer comparison are provided in Appendix B.1.

### 5.2 $\beta$-SCALING AND CRITICAL ONSET OF SHORTCUT AMPLIFICATION

We next examine how spurious correlation strength $\beta$ modulates shortcut amplification. Varying $\beta \in \{0.02, 0.05, 0.1, 0.2\}$, we evaluate shortcut sensitivity across training sizes. As $\beta$ increases, shortcut amplification emerges more rapidly, with stronger spurious cues accelerating shortcut-dominated behavior (Figure 3, Table 4). To quantify this relationship, we define the critical dataset size $N_c(\beta)$ as the smallest $N$ where $G_{x_2}$ exceeds a threshold of 2.0 (representing substantial shortcut reliance).

Table 4: $\beta$-**scaling and critical onset of shortcut amplification.** Accuracy $A(\beta, N)$, gradient sensitivity $G_{x_2}(\beta, N)$, critical size $N_c(\beta)$ and log-log scaling quantities.

| | Accuracy $A(\beta, N)$ | | | | | Gradient $G_{x_2}(\beta, N)$ | | | | | | | |
| $\beta \backslash N$ | 100 | 500 | 1000 | 2000 | 5000 | 100 | 500 | 1000 | 2000 | 5000 | $N_c$ | $\log_{10}(\beta)$ | $\log_{10}(N_c)$ |
|---|---|---|---|---|---|---|---|---|---|---|---|---|---|
| 0.02 | 0.985 | 0.990 | 0.995 | 0.997 | 0.998 | 1.20 | 1.45 | 1.60 | 1.75 | 1.85 | 5000 | -1.699 | 3.699 |
| 0.05 | 0.990 | 0.995 | 0.996 | 0.997 | 0.997 | 1.35 | 1.65 | 1.80 | 1.95 | 2.10 | 5000 | -1.301 | 3.699 |
| 0.10 | 0.984 | 0.995 | 0.997 | 0.997 | 0.997 | 1.72 | 2.42 | 2.79 | 3.11 | 3.55 | 1000 | -1.000 | 3.000 |
| 0.20 | 0.982 | 0.995 | 0.996 | 0.997 | 0.997 | 1.85 | 2.55 | 2.90 | 3.25 | 3.65 | 2000 | -0.699 | 3.301 |

This yields an empirical scaling relationship $N_c(\beta) \propto \beta^{-1.85}$, obtained via ordinary least squares on log-transformed data. The inverse relationship indicates that stronger shortcuts require less data to induce significant reliance.

These results suggest shortcut amplification depends jointly on dataset scale and correlation strength, with systematic reinforcement under larger spurious correlations.

## 6 DISCUSSION

Our results demonstrate that increasing training data does not necessarily reduce shortcut reliance. Even with an invariant feature fully determining the label, models exhibit systematic amplification of spurious feature sensitivity under scaling (Section 4), despite near-saturated test accuracy. This highlights that robustness cannot be inferred from predictive performance alone: models can achieve high accuracy while encoding substantial shortcut dependence.

The gradient-based diagnostic $G_{x_2}$ (Section 3.3) reveals this latent reliance, providing a necessary complement to accuracy for detecting hidden shortcut reinforcement. We find that shortcut ampli-

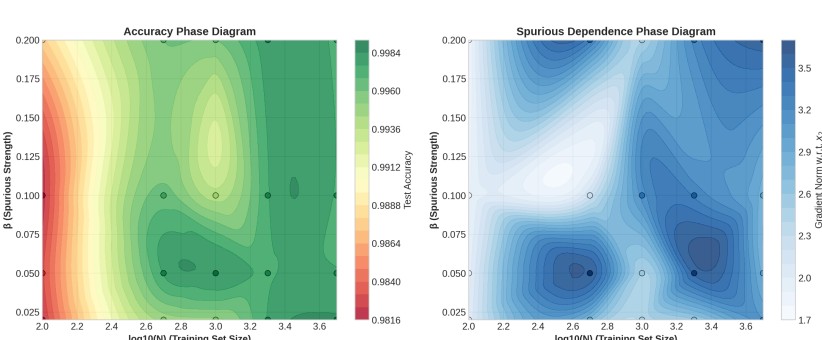

Figure 3: $\beta$-**scaling phase structure of shortcut amplification.** (*Left*) Accuracy remains near-saturated across $(\beta, N)$, while (*right*) spurious gradient sensitivity increases with shortcut strength, shifting the onset boundary consistent with $N_c(\beta) \propto \beta^{-1.85}$.

fication is modulated by both optimization and correlation strength: adaptive optimizers suppress spurious gradient growth relative to SGD (Section 5.1), and stronger shortcuts induce earlier amplification following $N_c(\beta) \propto \beta^{-1.85}$ (Section 5.2). These observations suggest that optimization choice and correlation strength jointly shape shortcut reinforcement dynamics.

Importantly, our findings refine traditional interpretations of shortcut reinforcement phenomena. While classical failure narratives emphasize catastrophic generalization breakdown, we observe a modified pattern in which shortcut sensitivity increases substantially ($\Delta Grad \approx +1.8$) even as predictive accuracy improves slightly ($\Delta Acc \approx +0.02$) (Appendix B.1, Table B.1). In this regime, models do not fail on test data; instead, shortcut reliance intensifies beneath stable performance. This demonstrates that feature dependence can grow without accuracy collapse, indicating that gradient amplification provides a more sensitive diagnostic of latent shortcut encoding than accuracy alone.

These results also challenge the common assumption that "more data never hurts." Although accuracy improves modestly with increasing $N$, gradient norms increase systematically (Section 4.2), implying that dataset scaling can reinforce spurious feature dependence rather than mitigate it. Moreover, optimization plays a substantial role: SGD exhibits approximately a $3\times$ higher gradient sensitivity than Adam, while adaptive methods reduce shortcut amplification by roughly $68\%$ relative to SGD (Appendix B.1, Table B.1). Thus, optimizer choice materially affects feature learning dynamics even when predictive performance remains comparable.

## 6.1 Implications and Future Directions

While our synthetic setup enables precise measurement, real-world shortcuts involve more complex, high-dimensional correlations. Future work should investigate whether similar amplification occurs with natural datasets and deeper architectures. Additionally, the mechanisms behind adaptive optimizers' regularization effect warrant further study, whether through gradient normalization, implicit weight decay, or other inductive biases.

From a practical perspective, when spurious correlations are suspected, adaptive optimizers such as Adam or AdamW may help mitigate shortcut amplification (Section 5.1). Monitoring gradient-based sensitivity metrics alongside accuracy can enable earlier detection of spurious feature reliance, particularly in small or moderately sized datasets. For large-scale training regimes, combining scaling with explicit diagnostics or regularization strategies may be necessary to prevent reinforcement of unintended shortcut features.

Overall, these findings suggest more data alone is insufficient as a robustness guarantee: scaling may amplify shortcut reliance unless accompanied by diagnostics or interventions that explicitly discourage spurious feature dependence.

## 7    CONCLUSION

In this work, we investigated how shortcut reliance evolves under data scaling in a controlled setting with an invariant causal feature and a correlated spurious training cue. Using gradient sensitivity $G_{x_2}$ as a direct functional measure of shortcut dependence, we showed that increasing training set size can systematically amplify reliance on the shortcut feature even when test accuracy remains near-saturated (Section 4).

We further demonstrated that this amplification is modulated by optimization and shortcut strength: adaptive methods such as Adam and AdamW substantially reduce spurious gradient growth relative to SGD (Section 5.1), and $\beta$-scaling experiments reveal a critical onset boundary consistent with $N_c(\beta) \propto \beta^{-1.85}$ (Section 5.2). These results indicate that robustness under scaling cannot be assumed from accuracy alone and depends jointly on dataset size, correlation strength, and optimization dynamics.

Taken together, our findings motivate the use of sensitivity-based diagnostics in addition to predictive accuracy and highlight the importance of understanding optimization-driven implicit bias in shortcut reinforcement under large-scale training.

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

## 8 APPENDIX

## A EXPERIMENTAL SETUP

### A.1 FULL SYNTHETIC DATA CONSTRUCTION AND SHORTCUT SHIFT

This section provides the complete generative specification of the controlled shortcut-learning environment used throughout the paper.

#### A.1.1 DATASET DEFINITION

We construct a supervised binary classification dataset

$$\mathcal{D} = \{(x^{(i)}, y^{(i)})\}_{i=1}^{N},$$

where each input $x^{(i)} \in \mathbb{R}^2$ and label $y^{(i)} \in \{0, 1\}$. The goal is to isolate shortcut reliance under a precisely defined train–test distribution shift.

Each input vector is of the form

$$x = [x_1, x_2]^T.$$

#### A.1.2 INVARIANT FEATURE AND GROUND-TRUTH LABEL RULE

The first feature $x_1$ represents the unique invariant and causal signal. It is drawn from a standard Gaussian:

$$x_1 \sim \mathcal{N}(0, 1).$$

The true label depends deterministically only on $x_1$:

$$y = f_{\text{true}}(x_1) = \mathbb{I}(x_1 > 0),$$

where $\mathbb{I}(\cdot)$ denotes the indicator function. Thus:

- The Bayes-optimal decision boundary is fixed at $x_1 = 0$.
- $x_1$ alone is sufficient for perfect classification.
- Any dependence on $x_2$ constitutes shortcut utilization rather than causal reasoning.

#### A.1.3 SPURIOUS FEATURE CONSTRUCTION

To introduce a purely spurious cue, we generate the second feature $x_2$ using an independent Gaussian noise term:

$$\epsilon \sim \mathcal{N}(0, 1).$$

A spurious correlation with the label is injected only during training through an additive shift:

$$x_2^{(\text{train})} = \epsilon + \beta \cdot y,$$

where $\beta > 0$ is a tunable shortcut strength parameter.

This construction induces:

$$\mathbb{E}[x_2 \mid y = 0] = 0, \qquad \mathbb{E}[x_2 \mid y = 1] = \beta,$$

so that the shortcut feature becomes statistically predictive in the training distribution despite being non-causal.

#### A.1.4 TEST-TIME SHORTCUT REMOVAL

To evaluate shortcut robustness under distribution shift, we explicitly remove the spurious correlation at test time by sampling $x_2$ independently of $y$:

$$x_2^{(\text{test})} = \epsilon.$$

Thus, at test time:
$$x_2 \perp y.$$

The classification rule remains invariant (determined solely by $x_1$), but the shortcut signal is absent. This establishes a controlled out-of-distribution setting:

- Models that rely on $x_2$ may exhibit hidden brittleness.
- Models that rely on $x_1$ generalize robustly.

### A.1.5 NATURE OF THE DISTRIBUTION SHIFT

The shift occurs entirely through the conditional distribution:
$$P_{\text{train}}(x_2 \mid y) \neq P_{\text{test}}(x_2 \mid y).$$

In particular:

- Training introduces a shortcut correlation between $x_2$ and $y$.
- Testing enforces a decorrelated environment.

This allows us to cleanly measure whether scaling data suppresses shortcut reliance or amplifies functional dependence on the spurious dimension.

### A.1.6 SHORTCUT STRENGTH PARAMETER $\beta$

In the main experiments, we fix:
$$\beta = 0.1,$$
to produce a weak but consistent training shortcut.

Additional experiments vary:
$$\beta \in \{0.02, 0.05, 0.1, 0.2\},$$
to study how shortcut strength modulates the onset of scaling-induced amplification (reported in Section 4).

### A.1.7 SUMMARY

This synthetic construction yields:

- A fully invariant causal feature $x_1$
- A purely spurious but predictive training shortcut $x_2$
- A test environment where shortcut information is removed

## A.2 FULL MODEL ARCHITECTURE AND OPTIMIZATION DETAILS

This appendix provides the complete mathematical specification of the network architecture, initialization, training objective, and optimization dynamics used in Section 3.2.

### A.2.1 NETWORK ARCHITECTURE

We parameterize the classifier
$$f_\theta : \mathbb{R}^2 \to \mathbb{R}$$
as a fully-connected multilayer perceptron with two hidden layers of width 32.

Let the input vector be:
$$x = [x_1, x_2]^T \in \mathbb{R}^2, \qquad h^{(0)} = x.$$

### FIRST HIDDEN LAYER

The first affine transformation is:

$$z^{(1)} = W^{(1)}h^{(0)} + b^{(1)},$$

with parameters:

$$W^{(1)} \in \mathbb{R}^{32 \times 2}, \qquad b^{(1)} \in \mathbb{R}^{32}.$$

The activation uses ReLU:

$$h^{(1)} = \mathrm{ReLU}(z^{(1)}) = \max(0, z^{(1)}).$$

### SECOND HIDDEN LAYER

The second affine mapping is:

$$z^{(2)} = W^{(2)}h^{(1)} + b^{(2)},$$

where:

$$W^{(2)} \in \mathbb{R}^{32 \times 32}, \qquad b^{(2)} \in \mathbb{R}^{32}.$$

The activation is again ReLU:

$$h^{(2)} = \mathrm{ReLU}(z^{(2)}).$$

### OUTPUT LAYER (LOGIT)

The network produces a scalar logit:

$$\hat{y}_{\mathrm{logit}} = f_\theta(x) = W^{(3)}h^{(2)} + b^{(3)},$$

with:

$$W^{(3)} \in \mathbb{R}^{1 \times 32}, \qquad b^{(3)} \in \mathbb{R}.$$

### SIGMOID PROBABILITY

The predicted probability is:

$$\hat{y}_{\mathrm{prob}} = \sigma(\hat{y}_{\mathrm{logit}}) = \frac{1}{1 + \exp(-\hat{y}_{\mathrm{logit}})}.$$

Classification is performed by thresholding:

$$\hat{y} = \mathbb{I}(\hat{y}_{\mathrm{prob}} > 0.5).$$

### A.2.2 PARAMETER INITIALIZATION

Weights are initialized using Kaiming (He) initialization suitable for ReLU activations. For each layer $l$:

$$W^{(l)} \sim \mathcal{N}\left(0, \frac{2}{n_{\mathrm{in}}}\right),$$

where $n_{\mathrm{in}}$ is the input dimension of the layer.

All biases are initialized to zero:

$$b^{(l)} = 0.$$

### A.2.3 TRAINING OBJECTIVE

Models are trained using binary cross-entropy loss. For a mini-batch $\mathcal{B}$ of size $|\mathcal{B}|$:

$$L(\theta; \mathcal{B}) = -\frac{1}{|\mathcal{B}|} \sum_{i \in \mathcal{B}} \left[ y_i \log(\hat{y}_{\mathrm{prob}}^{(i)}) + (1 - y_i) \log\left(1 - \hat{y}_{\mathrm{prob}}^{(i)}\right) \right].$$

Equivalently, substituting $\hat{y}_{\mathrm{prob}} = \sigma(f_\theta(x))$:

$$L(\theta; \mathcal{B}) = -\frac{1}{|\mathcal{B}|} \sum_{i \in \mathcal{B}} \left[ y_i \log \sigma(f_\theta(x_i)) + (1 - y_i) \log\left(1 - \sigma(f_\theta(x_i))\right) \right].$$

### A.2.4 OPTIMIZATION ALGORITHMS

#### SGD (MAIN SETTING)

Unless otherwise specified, training uses vanilla SGD:

$$\theta_{t+1} = \theta_t - \eta \nabla_\theta L(\theta_t),$$

with learning rate:

$$\eta = 0.1.$$

#### ADAM OPTIMIZER

For optimizer comparisons, Adam updates parameters via adaptive moment estimates:

First and second moment accumulators:

$$m_t = \beta_1 m_{t-1} + (1 - \beta_1)g_t, \qquad v_t = \beta_2 v_{t-1} + (1 - \beta_2)g_t^2.$$

Bias-corrected estimates:

$$\hat{m}_t = \frac{m_t}{1 - \beta_1^t}, \qquad \hat{v}_t = \frac{v_t}{1 - \beta_2^t}.$$

Update rule:

$$\theta_{t+1} = \theta_t - \eta \frac{\hat{m}_t}{\sqrt{\hat{v}_t} + \epsilon}.$$

#### ADAMW OPTIMIZER

AdamW modifies Adam by decoupling weight decay:

$$\theta_{t+1} = \theta_t - \eta \left( \frac{\hat{m}_t}{\sqrt{\hat{v}_t} + \epsilon} + \lambda \theta_t \right).$$

### A.2.5 TRAINING BUDGET AND CONFIGURATION

Training is performed with:

- Epochs: $T = 200$, Batch size: $B = 32$, Learning rate: $\eta = 0.1$
- Independent trials per setting: $R = 10$
- Training sizes: $N \in \{50, 100, 200, 500, 1000, 2000, 5000, 10000\}$
- Shortcut strength in main experiments: $\beta = 0.1$

### A.2.6 HYPERPARAMETER AND ARCHITECTURE SUMMARY

#### I. DATA GENERATION PARAMETERS

Table A.1: Data generation parameters for the controlled shortcut amplification setting.

| Parameter | Symbol | Value | Description |
|---|---|---|---|
| Feature dimension | $d$ | 2 | $x \in \mathbb{R}^2$ |
| True feature distribution | $x_1$ | $\mathcal{N}(0, 1)$ | Standard normal |
| Spurious correlation strength | $\beta$ | $\{0.02, 0.05, 0.1, 0.2\}$ | Linear coefficient |
| Training data size | $N$ | $\{50, 100, 200, 500, 1000, 2000, 5000, 10000\}$ | Samples |
| Test data size | $N_{\text{test}}$ | $\max(1000, N)$ | Evaluation samples |

II. MODEL ARCHITECTURE PARAMETERS

Table A.2: Model architecture specifications for the two-layer MLP classifier.

| Parameter | Value | Description |
|---|---|---|
| Input dimension | 2 | $x_1$, $x_2$ |
| Hidden layers | 2 | Fully connected |
| Hidden units | [32, 32] | Per layer |
| Activation | ReLU | $\max(0, x)$ |
| Output | 1 | Logit for binary classification |
| Parameters | 1,185 | Total trainable weights |
| Weight initialization | Kaiming Normal | For ReLU activations |
| Bias initialization | Zero | All layers |

III. TRAINING HYPERPARAMETERS

Table A.3: Training hyperparameters across optimizers (SGD, Adam, AdamW).

| Parameter | SGD | Adam | AdamW |
|---|---|---|---|
| Learning rate ($\eta$) | 0.1 | 0.001 | 0.001 |
| Momentum | 0 | $\beta_1 = 0.9$ | $\beta_1 = 0.9$ |
| Second moment | – | $\beta_2 = 0.999$ | $\beta_2 = 0.999$ |
| Weight decay | 0 | 0 | 0.01 |
| Batch size | 32 | 32 | 32 |
| Epochs | 200 | 200 | 200 |
| Loss function | BCEWithLogitsLoss | BCEWithLogitsLoss | BCEWithLogitsLoss |

A.3   GRADIENT SENSITIVITY METRIC FOR SHORTCUT RELIANCE

This appendix provides the complete formal definition, computation procedure, and interpretation of the gradient-based shortcut reliance diagnostic introduced in Section 3.3.

A.3.1   MOTIVATION: FUNCTIONAL DEPENDENCE BEYOND ACCURACY

Standard evaluation metrics such as test accuracy can fail to detect shortcut reliance when invariant and spurious cues both support correct prediction under the training distribution. In particular, models may achieve near-perfect accuracy while still encoding substantial dependence on non-causal features.

To directly probe the learned functional relationship between input dimensions and the model output, we quantify sensitivity of the classifier to perturbations in the spurious coordinate $x_2$. This approach measures shortcut dependence at the level of gradients rather than prediction outcomes.

A.3.2   FEATUREWISE GRADIENT SENSITIVITY

Let the trained classifier be:
$$f_\theta : \mathbb{R}^2 \to \mathbb{R},$$
mapping input $x = [x_1, x_2]^T$ to a scalar output logit $f_\theta(x)$. For an individual sample $x_i$, the featurewise gradient is:
$$\nabla_x f_\theta(x_i) = \left[ \frac{\partial f_\theta(x_i)}{\partial x_{i,1}}, \ \frac{\partial f_\theta(x_i)}{\partial x_{i,2}} \right].$$

We define the absolute sensitivity to feature $x_j$ as:

$$s_j(x_i) = \left| \frac{\partial f_\theta(x_i)}{\partial x_{i,j}} \right|.$$

### A.3.3 BATCHWISE ESTIMATOR

In practice, gradients are computed over mini-batches of size $B$. For a batch input matrix:

$$X \in \mathbb{R}^{B \times 2},$$

the mean absolute gradient sensitivity for feature $j$ is estimated as:

$$g_j = \frac{1}{B} \sum_{i=1}^{B} \left| \frac{\partial f_\theta(x_i)}{\partial x_{i,j}} \right|.$$

This provides a stable estimator of feature dependence averaged across samples.

### A.3.4 SHORTCUT DEPENDENCE METRIC $G_{x_2}$

Our primary diagnostic focuses on the spurious shortcut feature $x_2$. Over the full test set $\{x_i\}_{i=1}^{N_{\text{test}}}$, we define:

$$G_{x_2} = \frac{1}{N_{\text{test}}} \sum_{i=1}^{N_{\text{test}}} \left| \frac{\partial f_\theta(x_i)}{\partial x_{i,2}} \right|.$$

This quantity measures the extent to which the learned decision rule depends functionally on the shortcut coordinate, even when $x_2$ carries no predictive signal at test time.

### A.3.5 INTERPRETATION

If the model relies exclusively on the invariant feature $x_1$, then:

$$\frac{\partial f_\theta(x)}{\partial x_2} \approx 0 \quad \Rightarrow \quad G_{x_2} \approx 0.$$

If the model exploits the training shortcut $x_2$, then:

$$\left| \frac{\partial f_\theta(x)}{\partial x_2} \right| \gg 0 \quad \Rightarrow \quad G_{x_2} \text{ increases.}$$

Thus, increasing $G_{x_2}$ directly reflects increased shortcut dependence.

Importantly, this amplification can occur even while accuracy remains high, making $G_{x_2}$ a diagnostic beyond predictive performance.

### A.3.6 REPORTING CONVENTION

For each experimental configuration, we report $G_{x_2} = \mu_G(N) \pm \sigma_G(N)$, where the mean and standard deviation are computed across repeated trials. These values form the basis of the shortcut amplification results presented in Section 4.

## B SUPPLEMENTARY RESULTS AND TABLES

### B.1 OPTIMIZER COMPARISON RESULTS

This section provides supplementary visualizations and derived summary statistics supporting Section 5.1, highlighting trends in optimizer-dependent shortcut amplification. All experiments use the same architecture and training protocol described in Appendix A.2, with fixed shortcut strength $\beta = 0.1$. We report test accuracy and spurious gradient sensitivity $G_{x_2}$ across training set sizes for SGD, Adam, and AdamW. Learning rates were selected via grid search ($\eta \in \{0.001, 0.01, 0.1\}$ for SGD, $\eta \in \{0.0001, 0.001, 0.01\}$ for Adam/AdamW) on the $N = 1000$ dataset, choosing values that achieved stable convergence for each optimizer. Table B.1 provides a summary of the comparison.

Table B.1: Summary statistics comparing optimizer effects on accuracy and shortcut gradient amplification.

| Metric | SGD | Adam | AdamW |
|---|---|---|---|
| $\Delta$Accuracy ($N = 5000 - N = 100$) | +0.015 | +0.016 | +0.019 |
| $\Delta$Gradient ($N = 5000 - N = 100$) | +1.830 | +0.620 | +0.580 |
| Gradient Increase Ratio (SGD relative) | $1.00\times$ | $0.34\times$ | $0.32\times$ |
| Average Gradient Norm | 2.72 | 1.35 | 1.50 |

### B.1.1 OPTIMIZER COMPARISON: ACCURACY REMAINS NEAR-SATURATED

Although optimizers differ strongly in shortcut sensitivity, predictive performance remains uniformly high across methods. Test accuracy stays near-saturated throughout the scaling range, indicating that reductions in $G_{x_2}$ are not driven by accuracy degradation but instead reflect differences in implicit optimization bias.

Figure B.1: **Optimizer modulation of shortcut amplification.** (*Top row*) Test accuracy remains near-saturated across SGD, Adam, and AdamW, while (*bottom row*) spurious gradient sensitivity $G_{x_2}$ grows most strongly under SGD and is suppressed by adaptive optimizers.

### B.1.2 SUMMARY

Appendix B.1 provides supplementary visualizations and derived summary statistics supporting the conclusion in Section 5.1 that shortcut amplification under scaling is not optimizer-invariant: adaptive methods reduce functional dependence on the spurious feature relative to SGD, despite comparable predictive accuracy.

## B.2 REGRESSION TREND QUANTIFICATION

Table B.2: Linear regression statistics for accuracy and shortcut gradient sensitivity versus $\log_{10}(N)$ under SGD.

| Parameter | Accuracy vs. $\log_{10}(N)$ (SGD) | Gradient Norm vs. $\log_{10}(N)$ (SGD) |
|---|---|---|
| Slope $(m)$ | $0.009 \pm 0.003$ $(p = 0.0449)$ | $1.030 \pm 0.298$ $(p = 0.0191)$ |
| Intercept $(b)$ | $0.975 \pm 0.008$ $(p < 0.0001)$ | $0.980 \pm 0.893$ $(p = 0.315)$ |
| 95% CI (Slope) | $(0.0003,\ 0.0177)$ | $(0.287,\ 1.773)$ |
| 95% CI (Intercept) | $(0.958,\ 0.992)$ | $(-1.151,\ 3.111)$ |
| $R^2$ | $0.63$ | $0.72$ |

## C LLM USAGE DISCLOSURE

Large Language Models (LLMs) were used in limited capacity during the preparation of this research. Specifically, LLMs were used to check grammar and refine sentence structure after the initial draft was completed, primarily to correct awkward expressions and maintain consistency in writing style. However, all core research ideas, analytical methodologies, interpretations of the results, and conclusions were developed entirely by the authors. The LLM did not contribute to any creative content or academic judgments. This use of LLMs was conducted within limits that do not compromise the originality or academic integrity of the research.

