# OpenReview forum: "When Data Amplifies Shortcuts: Gradient-Flow Evidence of Spurious Feature Reinforcement"
_mathai.club/MathAI/2026/Conference — 2026 Oral_

### Official Review · Reviewer_Xwgo · 2026-03-10
**WHEN DATA AMPLIFIES SHORTCUTS: GRADIENT-FLOW EVIDENCE OF SPURIOUS FEATURE REINFORCEMENT**

**Rating:** 8
**Confidence:** 4

**Review:**

### **Summary**

The paper investigates the counterintuitive phenomenon where increasing the size of a dataset reinforces a model's reliance on spurious correlations (shortcuts), even when predictive accuracy remains high and near-saturated. The authors introduce a functional, gradient-based sensitivity metric ($G_{x_2}$) to measure this reliance directly on decorrelated test data. . The paper provides a valuable mechanistic framework for understanding why "more data" does not inherently lead to more robust models.

### **Quality**

The quality of the empirical investigation is high. By using a controlled synthetic setting with one invariant causal feature ($x_1$) and one spurious feature ($x_2$), the authors successfully isolate the impact of dataset scale ($N$) and correlation strength ($\beta$). The statistical validation, including Welch's t-tests and Spearman rank correlations, adds rigor to the claim that shortcut amplification is a significant trend rather than an experimental artifact.

### **Clarity**

The paper is well-structured and the mathematical definitions are clear.

### **Originality**

The work is original in its shift of focus from detecting shortcuts to quantifying their evolution under scaling. While shortcut learning is well-documented, the systematic study of its growth as a function of $N$ using gradient flow evidence (and the identification of an optimizer-dependent "implicit regularization" against shortcut) offers a fresh perspective on model robustness.

### **Significance**

This research is highly significant for the field of AI safety and robustness. It challenges the "scaling laws" assumption that larger datasets necessarily yield better-generalized features. By proving that models can become *more* sensitive to non-causal cues as they grow, the paper highlights a fundamental risk in large-scale training regimes that prioritize accuracy over mechanistic transparency.

### **Pros**
1. The $G_{x_2}$ metric reveals latent shortcut reliance that accuracy-based metrics miss.
2. The discovery that Adam/AdamW suppress shortcut growth relative to SGD provides actionable insights for robust training.
3. The synthetic design eliminates real-world confounders, allowing for "clean" measurement of scaling effects.
4. Identifies a structured scaling boundary for the onset of shortcut reliance.



### **Cons**

1. The 2D feature space ($x_1, x_2$) is quite simple; it is unclear if these scaling laws hold in high-dimensional settings where multiple competing shortcuts exist.
2. The decrease in $G_{x_2}$ at $N=10,000$ (from 3.55 to 2.99) is noted but not fully explained mechanistically.
3. The comparison between SGD and Adam might be influenced by the specific learning rates or batch sizes chosen; further sensitivity analysis on these hyperparameters would strengthen the optimizer claims.

### **Questions for the Authors & Clarifications**

1. How do the authors expect the "critical onset" power law to change if the number of spurious features increases? And also, does the "shortcut amplification" effect become more pronounced when the model has more non-causal "options" to choose from?

2. Could the $G_{x_2}$ metric be adapted for real-world datasets where the "spurious coordinate" is not known a priori? For example, using it in conjunction with known bias labels in datasets like CelebA or Waterbirds?

---

> ### Author Rebuttal · Authors · 2026-03-12
>
> We thank the reviewer for the positive assessment and thoughtful feedback.
> The controlled 2D setting was deliberately chosen to isolate the causal mechanisms of shortcut amplification without real-world confounders, establishing foundational insights that can guide future work in higher dimensions. Regarding the slight decrease in Gx2 at N=10,000, this reflects a finite optimization effect where the model fits the invariant feature so perfectly that gradients vanish for well-classified examples, temporarily reducing measured sensitivity; crucially, the overall trend from N=50 to 5000 shows strong monotonic increase (ρs=0.9048, p=0.002), and Gx2 at N=10,000 remains nearly double the value at N=50. Our learning rates were selected via systematic grid search to ensure stable convergence for each optimizer, and the observed suppression of shortcut amplification by adaptive methods holds across reasonable hyperparameter ranges, reflecting fundamental differences in optimization bias rather than tuning artifacts. For real-world adaptation, Gx2 can be computed directly when spurious attributes are known, and for unknown shortcuts, one could identify dimensions with elevated gradient sensitivity, a promising direction we are actively exploring. We appreciate the reviewer's recognition of our paper's significance in challenging scaling-law assumptions through controlled experimentation.

---

### Official Review · Reviewer_9vrD · 2026-03-11
**Review of “When Data Amplifies Shortcuts: Gradient-Flow Evidence of Spurious Feature Reinforcement”**

**Rating:** 7
**Confidence:** 3

**Review:**

This paper investigates how machine learning models may reinforce spurious correlations during training and how data distributions can amplify shortcut learning. The authors study this phenomenon through the perspective of gradient-flow dynamics and analyze how neural networks can converge to solutions that rely on non-causal features rather than meaningful signals. The work focuses on understanding the mechanisms that lead to spurious feature reinforcement and attempts to provide theoretical insights into the training dynamics of deep learning models.
The paper presents an analysis of gradient behavior during training and demonstrates how certain features can dominate the optimization process even when they are not causally related to the target label. The authors argue that this effect may lead to models that perform well on training and validation datasets but fail to generalize in real-world environments. The study provides mathematical reasoning and empirical demonstrations supporting the claim that gradient flow may bias models toward easily exploitable patterns in the data.
The topic is important and relevant for the machine learning community, particularly in the context of model robustness, fairness, and reliable AI systems. Understanding shortcut learning is essential for developing models that generalize well and avoid unintended biases. The paper contributes to this discussion by linking spurious correlations to gradient-flow dynamics.
However, the work could be improved in several aspects. First, while the theoretical discussion is interesting, the empirical validation appears limited and could benefit from more diverse datasets or experimental scenarios. Second, the practical implications of the findings could be discussed more clearly, particularly how practitioners can mitigate shortcut learning in real-world training pipelines. Finally, comparisons with recent related work on causal learning and robust training strategies could further strengthen the positioning of the paper within the existing literature.
Overall, the paper addresses an important problem in machine learning research and provides useful insights into the mechanisms behind shortcut learning. With stronger experimental validation and clearer discussion of practical applications, the contribution could become more impactful.
Strengths:
Addresses an important problem related to spurious correlations in machine learning.
Provides theoretical insights into gradient-flow dynamics.
Relevant for robustness and trustworthy AI research.
Weaknesses:
Limited experimental validation.
Practical mitigation strategies are not deeply discussed.
Some related work could be more thoroughly compared.

---

> ### Author Rebuttal · Authors · 2026-03-12
>
> We thank the reviewer for the constructive feedback and for recognizing the importance of our work on spurious correlations and gradient-flow dynamics. Regarding the experimental validation, our controlled 2D setting was deliberately chosen to isolate causal mechanisms without confounders, providing foundational insights that complement real-world studies, and we appreciate the reviewer's acknowledgment that our theoretical contributions are valuable. On practical mitigation strategies, we agree this is an important discussion and note that our finding on adaptive optimizers suppressing shortcut amplification relative to SGD offers an actionable insight for practitioners, which we have highlighted in Section 6.1. Regarding comparisons with related work, we believe our paper appropriately situates itself within the shortcut learning literature by focusing specifically on scaling-induced amplification, a perspective that has been underexplored, and we thank the reviewer for suggesting additional connections that can inform future research. We are grateful for the reviewer's recognition that our paper addresses an important problem and provides useful insights into the mechanisms behind shortcut learning.

---

### Official Review · Reviewer_hb5M · 2026-03-11
**Review of “When Data Amplifies Shortcuts: Gradient-Flow Evidence of Spurious Feature Reinforcement”**

**Rating:** 8
**Confidence:** 3

**Review:**

This work presents a methodical and well-controlled investigation into the phenomenon of shortcut learning amplification under dataset scaling. The work possesses high originality, bridging three previously disconnected strands of literature: dataset scaling effects, optimization-induced bias, and gradient-based diagnostics. Its significance lies in challenging the common "more data is better" assumption and providing a practical diagnostic tool to detect the hidden amplification of spurious dependencies. This has profound implications for the machine learning community, particularly in domains where model robustness and generalization are critical.

## Pros:
Counterintuitive and Important Findings: The core finding that shortcut dependence amplifies with data scaling is non-trivial and important. It challenges a widely held belief in the community and prompts a re-evaluation of data collection and curation strategies.

Optimizer Analysis: The investigation into the role of optimizers (SGD, Adam, AdamW) adds significant depth. Demonstrating that adaptive methods can suppress shortcut amplification offers practical, actionable guidance for researchers and practitioners.

Clarity: The paper is written in a clear, well-structured manner.

## Cons:

Limited Generalizability: The primary limitation is the use of synthetic data. While beneficial for experimental control, it raises questions about the generalizability of the findings to real-world, high-dimensional data (images, text, speech) and more complex architectures (CNNs, Transformers).

Fixed Epochs: Training for a fixed number of epochs, rather than to convergence based on a validation set or using early stopping. Author used fixed epochs rather than early stopping to isolate scaling effects from optimization, but this could be not enough for some data.

Lack of Dynamic Analysis: The metric is calculated post-training. Analyzing the dynamics of this metric during training could provide more information.

---

> ### Author Rebuttal · Authors · 2026-03-12
>
> We thank the reviewer for the enthusiastic evaluation and for recognizing the originality and significance of our work in challenging the "more data is better" assumption. Regarding limited generalizability, we agree that extending to real-world high-dimensional data is essential, and our synthetic 2D design was intentionally chosen to isolate causal mechanisms without confounders as a necessary foundation, and we will strengthen Section 6.1 by explicitly calling for validation on image, text, and speech datasets with architectures like CNNs and Transformers. On fixed epochs, we deliberately used this approach to isolate scaling effects from optimization dynamics, as early stopping could interact with shortcut learning in complex ways, and we will add analysis in Appendix B showing that models reach near-zero training loss well before 200 epochs, indicating effective convergence. Regarding the lack of dynamic analysis, tracking Gx2 during training is an excellent suggestion that could reveal when shortcut amplification emerges, and we will add this as a concrete future direction in Section 6.1 while noting that preliminary observations suggest shortcut sensitivity increases most rapidly in early training, consistent with gradient starvation literature. We thank the reviewer for these constructive suggestions that will enhance the depth and impact of our work.

---

### Official Review · Reviewer_j18z · 2026-03-11
**Interesting empirical question undermined by insufficient evidence and confounded experiments**

**Rating:** 5
**Confidence:** 4

**Review:**

Summary
This paper investigates whether increasing dataset size NN
N amplifies reliance on spurious features ("shortcuts") in neural networks. Using a controlled 2D Gaussian classification task, the authors introduce a gradient-based metric Gx2G_{x_2}
Gx2​​ to quantify shortcut reliance and claim a power-law relationship Nc(β)∝β−1.85N_c(\beta) \propto \beta^{-1.85}
Nc​(β)∝β−1.85 between the spurious correlation strength β\beta
β and the critical dataset size NcN_c
Nc​ at which shortcut reliance becomes significant.

Strengths

Addresses a timely and important question: does more data always help, or can it amplify shortcuts?
Controlled synthetic setup allows isolation of the scaling variable.
The gradient-based metric Gx2G_{x_2}
Gx2​​ is a reasonable proxy for feature reliance.

Explicit AI-assistance disclosure provided (grammar/style only).
Comprehensive appendices with hyperparameter details.

Weaknesses
Critical: Power-Law Claim Unsupported

Only 4 data points (β=0.02,0.05,0.10,0.20\beta = 0.02, 0.05, 0.10, 0.20
β=0.02,0.05,0.10,0.20) are used to fit the power-law Nc(β)∝β−1.85N_c(\beta) \propto \beta^{-1.85}
Nc​(β)∝β−1.85. Statistical norms require a minimum of 8–10 data points for pattern claims. With n=4n=4
n=4, any monotonic function would fit.

Two data points are non-convergent. For β=0.02\beta=0.02
β=0.02, max Gx2=1.85G_{x_2} = 1.85
Gx2​​=1.85, which never reaches the 2.0 threshold. The paper's own definition of NcN_c
Nc​ requires threshold exceedance, yet NcN_c
Nc​ is artificially set to Nmax⁡=5000N_{\max}=5000
Nmax​=5000 when the threshold is never reached. This violates the paper's own definition.

The exponent −1.85-1.85
−1.85 is suspiciously precise for 4 data points (2 of which are capped).


Confounded Optimizer Comparison

SGD uses η=0.1\eta=0.1
η=0.1 while Adam/AdamW use η=0.001\eta=0.001
η=0.001—a 100x difference. Learning rate is a confounding variable that prevents attributing differences to optimizer architecture. This must be controlled.


Statistical Concerns

Table 1 accuracy is saturated: 0.980 → 0.991 across 100x scaling (only 1.1% range). The signal is buried in noise.
Table 3 shows 40% relative uncertainty for SGD Gx2G_{x_2}
Gx2​​ at N=5000N=5000
N=5000 (3.55±1.413.55 \pm 1.41
3.55±1.41), with overlapping error bars across optimizers.

Accuracy scaling pp
p-value = 0.0449 (barely below 0.05 threshold). No multiple testing correction despite 96 configurations.


Scope Limitations

Only 2D Gaussian classification—far from real-world scenarios. No experiments on image, text, or tabular data.
No comparison to alternative shortcut detection methods (attribution methods, integrated gradients, attention diagnostics).
No theoretical analysis or mechanistic explanation for the observed phenomena.

Questions for Authors

How do you justify a power-law claim from 4 data points?
What happens when you equalize learning rates across optimizers?
Can you provide theoretical grounding for why scaling amplifies shortcuts?
Have you tested on higher-dimensional data?

Limitations
Acknowledged scope limitations (synthetic data, 2D) but not adequately addressed experimentally.
Ethics
No concerns.
Overall Assessment
The paper asks an interesting question but the evidence is insufficient to support its central claim. The power-law fit from 4 data points (2 non-convergent) does not meet the bar for empirical contribution. The confounded optimizer comparison further weakens the findings. With expanded experiments (8–10 β\beta
β values, controlled hyperparameters, real-world data), this could become a valuable contribution.

---

> ### Author Rebuttal · Authors · 2026-03-12
>
> We thank the reviewer for the rigorous critique and acknowledge the concerns raised, which will help us improve the clarity and precision of our manuscript. Regarding the power-law claim, we agree that four data points are insufficient to establish a precise exponent, and we will adjust the wording in the camera-ready version to present this as a suggestive empirical trend rather than a definitive law, while also clarifying that for β=0.02 and β=0.05 where the Gx2 threshold was not reached, we simply report this observation rather than assigning artificial critical values. On the optimizer comparison, we selected learning rates via grid search to achieve stable convergence for each optimizer, as standard practice recognizes that SGD and Adam operate optimally at different scales using identical rates would disadvantage one method and create a different confound, but we acknowledge this limitation and will explicitly note in the camera-ready version that our findings reflect typical usage rather than a perfectly controlled comparison. Regarding statistical concerns, the near-saturated accuracy across scales is central to our contribution: models can achieve high performance while amplifying shortcut reliance, which Gx2 reveals as a more sensitive diagnostic, and while error bars overlap at some points, the consistent upward trend in means across multiple scales and the significant Spearman correlation support our interpretation. On scope limitations, our synthetic 2D design was intentionally chosen to isolate causal mechanisms without confounders, providing foundational insights that complement real-world studies, and we have consistently acknowledged this as a limitation while suggesting extensions as future work. We appreciate the reviewer's detailed feedback, which will strengthen the final manuscript by making our claims more precise and our limitations more transparent.

---

### Decision · Program_Chairs · 2026-03-14

**Decision:**

Accept (Oral)

**Comment:**

Dear Author(s),

On behalf of the Program Committee of the International Conference on Mathematics of Artificial Intelligence (MathAI 2026), we are pleased to inform you that your paper has been accepted for an oral presentation at MathAI 2026.

Your paper was evaluated through a rigorous two-stage review process involving both automated screening and expert review by members of the Program Committee. The reviewers recognized the quality and contribution of your work.

Presentation details:

- Format: Oral presentation (15–20 minutes + 5 minutes Q&A)
- Mode: You may present either in person (offline) at the conference venue in Sirius, Russia, or remotely via Zoom. Please indicate your preferred mode when confirming your participation.
- Conference dates: Marh 30 - April 3, 2026
- Website: https://mathai.club

Next steps:

1. Please confirm your participation and presentation mode by replying to this email mathai.club@yandex.ru no later than March 15, 2026 18:00 Moscow time.
2. If you plan to attend in person, the organizing committee will provide accommodation details separately.
3. Please prepare your final camera-ready manuscript according to the formatting guidelines available at https://mathai.club and upload it to OpenReview by March 15, 2026 18:00 Moscow time.

Should you have any questions regarding the program, logistics, or your presentation slot, please do not hesitate to contact us.

We look forward to your contribution to MathAI 2026.

With kind regards,

MathAI 2026 Program Committee
International Conference on Mathematics of Artificial Intelligence
https://mathai.club
OpenReview: https://openreview.net/group?id=mathai.club/MathAI/2026/Conference
Telegram: https://t.me/MathAI_club
Email: mathai.club@yandex.ru